# Network Pharmacology of Adaptogens in the Assessment of Their Pleiotropic Therapeutic Activity

**DOI:** 10.3390/ph15091051

**Published:** 2022-08-25

**Authors:** Alexander Panossian, Thomas Efferth

**Affiliations:** 1Phytomed AB, Bofinkvagen 1, 31275 Vaxtorp, Sweden; 2EuroPharma USA Inc., Green Bay, WI 54311, USA; 3Department of Pharmaceutical Biology, Institute of Pharmaceutical and Biomedical Sciences, Johannes Gutenberg University, 55099 Mainz, Germany

**Keywords:** network pharmacology, adaptogens, gene expression, pleiotropic action, nonspecific, specific action

## Abstract

The reductionist concept, based on the ligand–receptor interaction, is not a suitable model for adaptogens, and herbal preparations affect multiple physiological functions, revealing polyvalent pharmacological activities, and are traditionally used in many conditions. This review, for the first time, provides a rationale for the pleiotropic therapeutic efficacy of adaptogens based on evidence from recent gene expression studies in target cells and where the network pharmacology and systems biology approaches were applied. The specific molecular targets and adaptive stress response signaling mechanisms involved in nonspecific modes of action of adaptogens are identified.

## 1. Introduction

Advantages of network pharmacology and the systems biology approach vs. the ligand–receptor-based reductionist concept were discussed recently in several reviews [1,2,3,4,5]. Pharmacology concerns drug action on physiological systems for therapeutic benefits, focusing on theories, procedures, and mechanisms related to the chemical control of physiological processes. Pharmacology aims to define the molecular events initiating drug effects at the pharmacological targets in therapeutic and other systems. The term “pharmacological target” refers to biomolecules, such as DNA, mRNA, and proteins, including transmembrane and nuclear receptors, ion channels, transport proteins, and numerous enzymes to which a drug binds first in the body to elicit its pharmacologic effect (Figure 1).

The complementary binding of drug molecules to a protein with a physiological purpose in the cell can change physiological response and result in a pharmacologic effect. The efficacy of addressing a drug (ligand) to its specific biomolecule (receptor) depends on the drug’s chemical structure and affinity.

Historically, the drug development strategy assumed that a single target mechanism of action is the best option to obtain the target-specific therapeutic, which is selective for treating specific conditions and free of adverse events. However, many drugs and natural compounds interact and bind with multiple receptors (multitarget interaction), resulting in polyvalent pharmacological action and pleiotropic therapeutic activity. For example, the polytropic therapeutic activity of ginseng and some natural compounds or “Ginseng-like” plant extracts, e.g., *Eleutherococcus senticosus* (Rupr. and Maxim.) Maxim., *Rhodiola rosea* L., *Withania somnifera* (L.) Dunal., *Schisandra chinensis* (Turcz.) Baill. *Andrographis paniculata* (Burm. f.) Nees, *Rhaponticum cartamoides* Iljin, and *Bryonia alba* L., collectively known as adaptogens, are associated with their polyvalent modes of action on the neuroendocrine-immune complex (stress system), multitarget effects on adaptive stress response signaling pathways, and molecular networks allied with these pharmacological targets [5,6,7,8,9]. Initially, adaptogens were defined as “Ginseng-like” plants, which increase the so-called “state of nonspecific resistance” of an organism to stress, resulting in tonic, stress-protective, and adaptogenic activity [5]. Adaptogens are currently defined as a therapeutic category/pharmacological group of herbal medicines or/and nutritional products, increasing adaptability, survival, and resilience in stress and aging by triggering intracellular and extracellular adaptive signaling pathways of cellular and organismal defense systems (stress system, e.g., neuroendocrine-immune complex). Furthermore, adaptogens trigger the generation of hormones (cortisol, corticotropin-releasing hormone (CRH), gonadotropin-releasing hormones, urocortin, neuropeptide Y), playing key roles in metabolic regulation and homeostasis [9].

Our recent studies revealed the genome-wide effects of several adaptogenic herbal extracts in brain cell culture [6,7,8,10,11,12,13,14,15]. These data highlight the consistent activation of adaptive stress response signaling pathways (ASRSPs) by adaptogens in T98G neuroglia cells [6]. The adaptogens affected many genes playing critical roles in the modulation of adaptive homeostasis, indicating their ability to modify gene expression to prevent stress-induced and aging-related disorders [9]. These studies provided a comprehensive look at the molecular mechanisms by which adaptogens exert stress-protective effects.

However, the rationale for the pleiotropic therapeutic efficacy of adaptogens and the specific molecular targets and adaptive stress response signaling mechanisms involved in nonspecific modes of action of adaptogens have not been clearly defined.

## 2. Trends in Network Pharmacology Research

### 2.1. On the Road to Network-Based Precision Medicine

The advent of systems biology led to a paradigm change in medicine from symptom-based diagnosis and therapy to individualized therapy, where drug treatments are based on the individual disease parameters of each individual patient. The expectations are that individualized therapy is more precise and would lead to improved treatment outcomes. Hence, the term “precision medicine” has been proposed. Developing techniques that allow monitoring the expression of genes, proteins, or metabolites in their entirety in cells, tissues, and organs was the basis for a new discipline, systems biology. These comprising datasets of biomolecules are considered to better reflect disease-related pheno- and genotypes than conventional methods in biomedicine. The huge expression profiles derived from these molecules cannot be inspected by the eye and have to be evaluated with the help of bioinformatical tools to grasp the relevant data from a large background of irrelevant information to reconstruct meaningful networks from complex biomolecular interactions. The aim of “network medicine” is to understand the pathophysiology of diseases at the systems level. Disease-related interaction networks (“interactomes”) form the basis for the development of novel drugs that address networks rather than single targets. This is the aim of “network pharmacology” and “network medicine” [1,16].

### 2.2. Methods of Network Pharmacology

During the past two decades, we have been facing a technical revolution with novel methods that collectively have been termed “omics” technologies. The term “omics” is derived from the Latin suffix “-ome” and indicates mass measurements in an endeavor to understand a biological question or phenomenon in an integrated manner [17,18]. Typical omics technologies comprise genomics, transcriptomics, proteomics, and metabolomics. More recently, further methods came about, such as epigenomics, lipidomics, glycomics, etc. Micro-RNA and long non-coding (lnc) RNA profiles, as well as microbiomes, are also counted as “omics” technologies. The microbiomes in the gut, skin, and other organs have been considered the “extended genome” from microorganisms that add to the human genome [19].

#### 2.2.1. Genomics

Sequencing of the whole genome or exons provides information on single nucleotide polymorphisms and copy number variations (deletions, duplications, amplification). Genome-wide association studies (GWASs) have been used to identify profiles and markers for a wide variety of human diseases [20,21]. The challenges associated with GWASs are the elucidation of modes of action related to these genetic variants since (1) the function of the affected genes is not always well understood, (2) it remains challenging to distinguish between causative and concomitant changes without function, and (3) the functional relevance of genetic variations in non-coding regions of the genome is frequently unknown [22].

#### 2.2.2. Transcriptomics

Transcription and translation are biological key processes in both physiological and pathophysiological conditions. Comprehensive determination of all RNA transcript expression profiles supports the understanding of biological processes in diseases and intervention by medications [20]. Transcriptomic studies are mainly based on microarray hybridization or RNA sequencing and help to elucidate not only novel pathways of diseases but beyond the identification of novel networks of interacting genes that are much more complex than straightforward signal transduction pathways. Transcriptomics is of immense value in exploring changes in expression during disease progression and monitoring the success (or failure) of drug treatment. 

A surprising finding, at first sight, might be that there is only a modest correlation between transcriptomics and proteomic expressions [20,23,24]. This may indicate not only differences in the robustness of quantification methods between RNA and protein methods but may also reflect real biological situations, e.g., mRNA and protein expression of the same gene may occur in temporal distances and additional post-translational regulatory mechanisms (e.g., by miRNAs) [20].

#### 2.2.3. Proteomics

The proteomic method involves the expressions of all proteins, including isoforms (e.g., of enzymes), amino acid mutations, and posttranslational modifications (e.g., phosphorylation, prenylation, acetylation, etc.). All these variants may affect the functions of proteins and peptides. Hence, proteomic results reflect not only the expression status of a protein but also allow insights into its functional status in healthy vs. diseases tissues, treated vs. untreated cells, etc. [25,26].

#### 2.2.4. Metabolomics

Mass spectrometry and nuclear magnetic resonance-based protocols allow the profiling of all low molecule weight compounds in tissues, cells, body fluids, and other biological samples [27,28]. It links chemical profiles with other “omics”-derived profiles in a broader sense to entire cellular and organismic “footprints”.

#### 2.2.5. Other “Omics” Technologies

Lipidomics: Lipids play a crucial role, not only in the cellular architecture (e.g., as lipid bilayers in biomembranes) but also in signal transduction processes. Therefore, the large-scale measurement of lipids enlarges our understanding of signal networks in pathophysiology and drug treatment monitoring [29,30].

Glycomics determines the presence of sugars (free or complexed with other molecules like glycolipids and glycoproteins) [31].

Microbiomics: An emerging concept in network pharmacology is that microorganisms in the gut, skin, and other organs are relevant mediators of inflammation and other organismic processes. Thus, microbial genomes can contribute to pathogenesis [32].

### 2.3. “Multi-Omics” Technologies

It can be expected that the synopsis of several “omics” methods will improve understanding of the complex regulatory circuits of cells and tissues compared to single “omics” techniques. Therefore, “multi-omics” approaches attempt to integrate data from genomics, transcriptomics, proteomics, metabolomics, etc. in a concerted manner. Bioinformatical integration of multi-omics data can be performed by unsupervised (e.g., correlation-based Bayesian methods) or supervised methods (e.g., multiple kernel learning) [22,33,34]. Multi-dimensional networks can provide much more precise knowledge of the expression dynamics of disease-related profiles. This may refer to a holistic aspect of network pharmacology. A specific challenge in “multi-omics” is that datasets are not always complete, and missing data points engrave reliable model construction by bioinformatics.

### 2.4. Single-Cell “-Omics” Technologies

The above chapters described the application of “omics” technologies under the assumption that the investigated tissues are homogeneous, which might frequently not be the case. While cell lines might be more homogenous than healthy tissues, the complexity and heterogeneity in diseased tissues are even much higher. Therefore, novel techniques have emerged to analyze gene expression at the level of single cells. Spatial transcriptomics allows to simultaneously identify the diverse location-specific expression of single isolated cells. In comparison to the investigation of bulk tissue samples, single-cell techniques lead to more accurate transcriptome profiling [22,35]. Single-cell “omics” requires the isolation of cells either by flow-activated cell sorting or by microdissection microscopy. The analysis of single cells is not restricted to spatially separated areas, such as cells from the central and peripheral nerve system or cells of different histological origins within a tumor (e.g., epidermal cancer cells, fibroblasts, immune cells).

Another dimension that can be added to this kind of analysis is time. Temporal expression profiles of tissues may significantly differ. Illustrative examples are (1) the different developmental phases from an embryo to an adult, (2) the stress response of tissues upon exposure to xenobiotics, (3) the treatment response of diseased tissues upon drug therapy, and (4) the early detection of relapsing cancer cells after chemotherapy (minimal residual disease) [36,37]. Despite the attractiveness of single-cell technologies, their shortcomings should not be concealed, such as high costs and low coverage [22].

### 2.5. Network Modeling

Disease-associated genetic variants and expression profiles of complex phenotypes require integrated bioinformatical models to extract useful information from huge amounts of data. Visual inspection without bioinformatical tools will barely enable a better understanding of the pathophysiology of a disease or the prediction of drug response for individual patients. This has been described as a “big data problem” [29]. Hence, elaborated bioinformatical modeling approaches have been reported during the past years. One strategy is the “top-down” approach, which assigns identified genes to disease-related networks. The “bottom-up” approach takes advantage of existing knowledge on gene functions, biological relationships, and network connections to create novel disease-related molecular networks [38]. Both approaches suffer from existing knowledge gaps regarding the incompleteness of the molecular interactomes, defining key genes as “nodes” in the networks, etc. Hence, the biological relevance of identified molecular networks needs to be verified by corresponding in vitro and in vivo experimentation. There is currently a thriving advancement in methods from artificial intelligence, bringing up more sophisticated models to calculate precise molecular networks. In the context of network pharmacology, this is especially promising since classical pharmacology focuses on single transduction pathways and the effects of drugs on them. In system biology, novel networks of “nodes” (functional central elements) and “edges” (functional interactions) appear. These networks are usually much more complex than single signal transduction pathways. They may contain several of these signal transduction pathways (“modules”) and additional interactive relationships between them that have not been known before.

It is common sense in biomedicine that genes cooperate in interconnected networks rather than linear pathways [39,40]. Thus, co-expression networks can be constructed between control and test samples to identify closely related nodes and modules. However, these networks may differ between comparison pairs, i.e., healthy vs. diseased tissues, drug-treated vs. untreated cells, etc. Comparing paired samples facilitates finding those interactomes in the test samples that most significantly differ from the control samples [40,41].

Another question is how to filter out relevant from a bulk of irrelevant information from larger datasets with many patients. Two approaches have been suggested: The heterogeneity of large sample groups (e.g., caused by confounding factors, batch effects, etc.) may be overcome by the “reductionist” approach, which matches closely related control and test samples (e.g., placebo and verum receiving patients in clinical trials) and eliminate non-matching samples. The other possibility is the “holistic” approach, which integrates as many datasets as possible and subjects them to statistical models that consider large sample variability [40,42].

A further aspect that needs to be considered for modeling network interactions is whether the biological systems are adaptive or non-adaptive. Non-adapting systems are relatively stable over time (e.g., a progressing disease) compared to adaptive systems that can rapidly change upon appropriate stimulation (e.g., the immune system) [43].

Another frequently neglected point is that the main emphasis thus far has been put on the structure of molecular networks (i.e., which gene interacts with which one) but not on the magnitude of interactions between nodes and modules and also without consideration of diversity caused by temporal and spatial differing interactions [43].

The need to process huge amounts of “omics”-generated data fostered applications from machine learning and artificial intelligence [44,45,46]. Artificial intelligence enables the evaluation of huge datasets to extract the most relevant information and detect otherwise hidden information. Neural network algorithms require sufficient high computing power to classify complex datasets. This is especially true for high-dimensional “multi-omics” data [22].

Processing huge multi-dimensional datasets bears the danger that the generated models are biased because of overfitting, multi-collinearity of parameters, and infinite coefficient solutions [47]. Specific dimension-reducing algorithms (e.g., random forest, support vector machines, singular value decomposition) are frequently applied to avoid these issues.

The overall aim of “omics”-based disease modeling is to develop tools coming up with well-characterized molecular pathways that enable building reliable interaction networks and exploring the basic drivers of disease [38,48]. If drugs for these disease drivers are developed, we will come closer to what has been termed “network medicine” [49,50]. Merging network medicine with individualized precision medicine can potentially be a game-changer in pharmacological research.

### 2.6. “Omics”-Based Biomarker Development

Biomarkers are valuable in molecular medicine for the refined subtyping of diseases, as indicators of disease progression, for the selection of targeted therapies, as predictors of drug response, and as prognostic markers for the survival of patients. Hence, it is no surprise that the profiles generated using “omics” technologies have also been explored for their value as biomarkers. The task of identifying “omics”-based biomarkers is not trivial, not only because biomarker profiles consist of several or many genes but also because the genetic and regulatory networks may considerably differ from patient to patient [51]. Individually varying disease marker profiles may be explained by individually different lifestyles (environmental factors, comorbidity, comedication, individual habits, etc.) [52]. One of the first commercial kits is a proof-of-principle that “omics” technologies can be used for the development of biomarkers. The DNA microarray-based diagnostic kit MammaPrint is based on 70 genes to determine the prognosis of breast cancer patients [53].

The influence of environmental factors should not be underestimated since polymorphisms in xenobiotic-ad drug-metabolizing enzymes represent a large factor influencing not only drug response and treatment outcome but also drug–drug interactions and adverse side effects [52]. The greatest challenge in “omics”-based biomarker development is, however, to delineate reliable biomarkers to select the right drug for the right patient.

### 2.7. “Omics”-Based Therapy Monitoring

Among the primary goals is the modeling of diseases in general and the generation of models at different stages. Progressed diseases are frequently not well treatable by drugs. Thus, we need models of disease progression. One strategy is to generate multiple turnover models that are connected with each other in a cascading fashion [43]. Sequential disease models better reflect the complexity of diseases and mediate a more holistic perspective for network pharmacology. It can be imagined that different drugs for different stages of the same disease could be developed. Complexity is certainly one of the main problems, and it is essential to reduce the number of parameters in huge datasets without worsening the dynamic behavior of the model [54,55].

In classical pharmacology, it has been the primary goal to develop mono-specific or at least highly specific drugs. This concept was difficult to realize since most drugs are more or less promiscuous. Having the complexity of “omics”-based network models in mind, systems biological interventions may take advantage of this fact, and novel drugs could be developed that are capable of targeting multiple selected targets at the same time [56,57,58] (Figure 1). A proof-of-principle of this concept represents several clinically approved multi-kinase inhibitors [59,60]. The realization that mono-specificity is more fiction than fact in classical pharmacology also led to the “drug repurposing” concept. Old drugs that are approved for a certain disease indication occasionally reveal activity in other diseases, too [61,62,63,64,65]. Novel network-based methods of studying drug–target interactions and their disease–gene relationships represent attractive tools to systematically screen the human protein–protein interactome to reposition already approved drugs [66,67].

Moreover, this concept substantiates the hypothesis that network pharmacology is powerful in explaining the multi-specific activity of many—if not all—pharmacologically active natural products. In this sense, network pharmacology makes, for the first time in pharmacological research, the use of complex herbal mixtures in phytotherapy more rational. We will elaborate on this point in more detail below.

### 2.8. Network Pharmacology with a Natural Product and Complex Herbal Mixtures

As outlined above, former drug discovery programs focus on potent drugs that exert (some) specificity to a given therapeutic target. Due to the promiscuity of many drugs to modulate multiple targets and pathways [56,68], the term “polypharmacology” has been coined. Bioinformatical tools have the power to design multi-specific drugs [69,70,71]. Thus, drug promiscuity turned from the appearance of unwanted effects to an intended strategy in drug discovery and development. Network pharmacology can now integrate complex “omics”-based datasets from “multi-omics” and single-cell “omics” technologies to form the fundament for novel next-generation drugs. Intriguingly medicinal plants ad phytochemicals play a central role in this emerging new field of drug research.

Botanicals and phytochemicals are indispensable sources for medications both in traditional medicines and classical drug development [72,73]. Natural products possess tremendous structural diversity and are characterized by their target-related multi-specificity [74]. Thus, they can be understood as examples of natural polypharmacology. The construction of molecular networks for compounds isolated from medicinal plants and complex herbal mixtures has emerged as a thriving new field in network pharmacology [73]. Among numerous approaches accumulated in the literature, we choose two of them to exemplarily illustrate the multiple-site attacks of natural products in cancer therapy and the usefulness of network construction to unravel these bioactivities: (1) Natural products can simultaneously target several pathways in a tumor and (2) they can modulate the immune response in the tumor microenvironment [75,76]. The main obstacle to successful chemotherapy of tumors is the development of drug resistance caused by the heterogeneity of tumor subpopulations [77]. Complex herbal mixtures represent smart combination therapies of dozens (or more) of chemical ingredients that decrease the risk of resistance development and exert synergistic interactions. Network construction facilitates the elucidation of multiple targets for natural products and the observation of changes in the network profiles as early predictors of therapy response in individual patients. In principle, this concept can be transferred to many other diseases too.

In general, network pharmacology can be applied to all kinds of traditional medicines. However, most progress has been made by applying systems biology-based pharmacology to traditional Chinese medicine (TCM) [78,79]. Network pharmacology seems to fit into the holistic philosophy of TCM perfectly. While there is a plethora of clinical data reporting the efficacy of TCM, the mechanistic understanding of herbal TCM formulae is still in its infancy [80,81,82,83,84]. The expectation is that the theory of TCM, which fundamentally differs from the reductionistic concept of western medicine, could be scientifically explained [85,86,87]. It is hoped that network-based multitarget-multicomponent models help to understand the characteristic TCM syndromes [88] rationally. A syndrome in TCM describes an imbalance in the body that could lead to different diseases, e.g., hot or cold syndromes cause hot- or cold-related diseases [89].

### 2.9. Toxicology

Drugs (synthetic or natural) are xenobiotics, and the organism tries to eliminate them. While pharmacokinetic models describe the elimination of medications to estimate the beneficial pharmacological effects, the metabolism of drugs can also provoke toxic reactions. This is true, especially in cases where individual variations in the xenobiotic metabolism lead to overshooting reactions, which frequently manifest in hepato- or nephrotoxicity [90]. The xenobiotic metabolism is not only influenced by endogenous factors, such as single nucleotide polymorphisms in drug-metabolizing enzymes but also by exogenous factors, such as smoking and alcohol consumption, other lifestyle habits, environmental pollution, etc. 

In some cases, polypharmacological medications, including natural products and medicinal plants, can increase the risk of side effects and even severe toxicities. This phenomenon is already known for decades, but novel techniques enable the generation of “toxicity-gene-targeting-drug” networks to further improve safety predictions for natural products and botanicals [91]. Toxicogenomics is an area of research that explores toxic reactions of xenobiotic compounds at the “omics” level, which is thought to be more sensitive than conventional methods [18,92,93]. Hence, toxicogenomic approaches may serve as early toxicity predictors [94]. Therefore, it is highly recommendable to supplement network pharmacology with suitable approaches of network toxicology to assure the safe consumption of phytotherapy. 

## 3. Specific and Nonspecific Actions of Adaptogens

Extensive research of herbal medicines in the past decades has provided more evidence that, as a rule, they exhibit polyvalent nonspecific pharmacological activity affecting many physiological functions and regulatory systems in humans, e.g., *Panax ginseng*, *Andrographis paniculata*, *Withania somnifera*, *Curcuma longa*, etc. [5,95,96,97,98,99,100,101,102,103,104,105]. Some of them, e.g., *Panax ginseng*, *Bryonia alba*, etc., were traditionally used as panaceas [95,105]. A rational explanation for their mysterious pleiotropic actions remains a challenge. 

Depending on the chemical compositions of herbal extracts, the pharmacological effects of certain medicinal plants are specific to some extent, e.g., *Hypericum perforatum* is known mainly as an anti-depressant [106], but in the meantime, it exhibits a variety of other pharmacological activities, such as antiviral, antitumor, anti-stress, etc. [106,107,108,109].

In the past century, Brekhman and Dardimov suggested that some of the “Ginseng-like” plants can increase the so-called “state of nonspecific resistance” of an organism to stress, resulting in tonic, stress-protective, and adaptogenic activity [110]. Regretfully, some researchers misinterpret that definition of adaptogens as agents, which does not reveal any specific therapeutic action. Others freely declare some plants [111] or natural substances [112] as adaptogenic without scientific evidence of their efficacy and safety and lack of knowledge of the mode or mechanisms of action.

Meanwhile, the modes and mechanisms of action of adaptogens have been studied for seven decades [5,9,113] and elucidated to some extent due to the implementations of methods of molecular biology, network pharmacology, and systems biology concepts [6,7,8,10,11,12,13,14,15].

The similarity of chemical structures of ginsenosides with the stress hormone cortisol (Figure 2), suggests that their mechanism of action is associated with glucocorticoid receptors and the mode of action with the hypothalamus–pituitary–adrenal (HPA) axis—a functional part of the neuroendocrine-immune complex, collectively known as a “stress system” [114,115], which regulates adaptability, survival, and resilience of organisms in stress and progression of aging-related disorders [96,116], including neurodegenerative diseases (Alzheimer’s disease, Parkinson’s disease, senile dementia, etc.), atherosclerosis, cardiovascular disease, metabolic diseases (type 2 diabetes, obesity, and hypertension) [117], muscle degeneration (sarcopenia), degenerative joint disease (osteoarthritis), cancer, etc.

Indeed, in the earlier studies dated 1970–1980, it was demonstrated that ginsenosides act as functional ligands of glucocorticoid receptors [119,120,121]. This finding provides a rationale for the pleiotropic pharmacological activity of ginseng and its promising efficacy in numerous diseases associated, which chronically increased (immune suppression, melancholic depression, increased arousal or anxiety, loss of libido, suppression of feeding/anorexia, gastrointestinal dysfunction, increased blood pressure, tachycardia, chronic active alcoholism, alcohol and narcotic withdrawal, etc.) and decreased (chronic fatigue, somnolence, decreased arousal and performance of the task, fibromyalgia syndromes, increase in appetite, and weight gain, etc.) the level of cortisol in the blood circulation system. 

Further findings revealed the mechanisms and action of ginsenosides are associated with many other molecular targets except glucocorticoid receptors and multiple modes of action related to the neuroendocrine-immune complex and other regulatory systems involved in maintaining homeostasis and survival. Figure 3 shows the modes of the pharmacological action of red ginseng, describing functional changes of cells, physiological and regulatory systems involved in defense response at various levels of regulation of homeostasis, and the phases of progression of diseases [113,114,115,116,118]. 

The ginsenosides act primarily on the hypothalamus and pituitary, stimulating ACTH secretion, followed by increased corticosterone biosynthesis in the adrenal cortex [117,119,120,121]. On the contrary, ginseng has an inhibitory effect on the hyperactivity of the HPA axis induced by stresses and increased corticosterone levels associated with metabolic and psychiatric disorders, e.g., Ginsenoside Rd, inhibits corticosterone secretion in the cells, and inhibits ACTH-induced corticosterone biosynthesis through downregulation of proteins in the cAMP/PKA/CREB signaling pathway in adrenocortical cells [122] (Figure 3). In other words, ginseng acts as a mild stressor (“stress vaccine”), increasing the range of adaptive homeostasis that adjusts the stress response in mental disorders and metabolic diseases. That is a typical adaptogenic activity to activate the body’s defense system and metabolic rate resulting in increased resilience and survival in response to stressful factors, including infections [9]. Key mechanisms of action of ginseng and other adaptogens are related to their effects on adaptive intracellular signaling pathways [6,7,8], specifically, PI3K-AKT/PkB-Nrf2 [103,123,124,125,126], stress/mitogen-activated protein kinase (SAPK/MAPK)-, JNK(MAPK13)-, p38(MAPK10)- [125,126,127], and ERK-mediated signaling pathways involved in the regulation of cell growth, differentiation, apoptosis, and survival under the stressful stimulus, factors including, hormones, neurotransmitters, xenobiotics, pathogens, and physical factors (UV, osmotic, etc.) (Figure 3).

Interactions of ginsenosides with various cellular targets, including key enzymes, transcription factors, and plasmatic and nuclear receptor proteins involved in these signaling pathways, result in modified cellular responses and different therapeutic actions depending on the target cell type.

In neurons and neuroglia cells, ginseng-induced activation of AKT signaling and inhibition of JNK results in neuroprotection, neurogenesis, proliferation, cells survival, prevention of progression of aging-related neurodegenerative diseases (Alzheimer’s, Parkinson’s), long-term potentiating, improvement of memory learning, attention deficit, cognitive functions, mental performance and fatigue, and beneficial effects in behavioral and mood diseases (anxiety, depression) Figure 3.

The classical reductionist model that presumes a specific receptor/drug interaction is not suitable for understanding the molecular mechanisms of action of adaptogens associated with the physiological notion of “adaptability” [9]. On the contrary, the systems biology and network pharmacology concepts provide ideal mechanistic tools for understanding and conceptualizing adaptogen modes and mechanisms of action. Adaptation to environmental challenges and senescence are multistep processes that involve diverse mechanisms and molecular interactions. Many molecular networks regulate and harmonize intracellular and extracellular communications, metabolic regulation of homeostasis of various cells, and physiological systems.

Multiple molecular targets of adaptogens, molecular networks, and adaptive stress response signaling pathways have recently been identified [5,6,7,8,9,10,11,12,13,14,15]. They are associated with chronic inflammation, atherosclerosis, neurodegenerative cognitive impairment, metabolic disorders, and cancer, which are more common with age [9].

Overall, adaptogens trigger ***pleiotropic*** genes, molecular mechanisms, and cellular signaling pathways that mediate adaptive and defense responses, resulting in multitarget modes of action simultaneously and, therefore, in ***nonspecific pleiotropic*** pharmacological activity.

Pleiotropy is the result of the effect of adaptogen on a single gene that impacts multiple signaling pathways, biological processes, physiological functions, and phenotype characteristics. Various cells use the gene transcription mechanism that triggers numerous downstream signaling pathways and molecular networks that collectively affect multiple molecular targets, resulting in many pharmacological activities (nonspecific effect).

Pharmacological activities of adaptogens (Table 1) depend on their ***molecular mechanisms of action***, including ***specific*** effects on the expressions of genes (Table 2), encoding proteins of adaptive stress response *signaling pathways* (Table 3), and networks involved in ***the modes of the pharmacological action***, including the regulation of biological processes and physiological and cellular functions (Table 4), which are associated with progression of stress-induced and aging-related diseases (Table 5) as depicted on the flowchart below, Figure 4.

***The mechanisms of action*** of adaptogens describe the molecular changes and their extracellular and intracellular interactions. Figure 5 shows an example of a molecular network and interactions associated with predicted inhibition of exocrine gland tumor by an extract of red ginseng.

***The modes of the pharmacological action*** of adaptogens describe functional changes of cells and regulatory systems involved in defense response at various levels of regulation of homeostasis and the phases of progression of diseases [128]. 

Table 1, Table 2, Table 3, Table 4 and Table 5 show predicted and evidence-based health claims and indications for the therapeutic use of adaptogens in various diseases, pharmacological activities of adaptogens, their effects on biological processes, physiological and cellular functions, canonical signaling pathways, and several essential genes triggering the effects of adaptogens. 

There are several limitations of network pharmacology predictions based on in vitro and in silico studies, which must be further verified in animal experiments. The second limitation is the limited scientific information related to the direction of correlations between gene expression and physiological function or disease that is used in silico analysis for predictions of therapeutic efficacy or toxicity. Additional studies where different experimental outcome measures will be applied are required. Furthermore, a strong resistance mechanism can override all other effects of a drug. For instance, the drug efflux transporter *p*-glycoprotein in the cell membrane can expel drugs before they can reach their actual intracellular targets and thereby prevent all or most downstream signaling and network effects. Lastly, clinical studies on predicted diseases in human subjects are essential.

There are no limitations concerning pharmaceutical-grade herbal extracts of reproducible quality and pharmacological activity compared to purified active constituents since the content of a genuine extract is adjusted to the defined content or range of the active constituents with known therapeutic efficacy or to the content of the analytical markers.

Purified compounds can also act as adaptogens [129,130] if there is evidence that they increase adaptability, survival, and resilience in stress and aging by triggering intracellular and extracellular adaptive signaling pathways of cellular and organismal defense systems, e.g., the generation of hormones (cortisol, corticotropin-releasing hormone, and gonadotropin-releasing hormones, urocortin, neuropeptide Y [9]. Currently, at least three purified adaptogens, ginsenoside Rg1, salidroside (rhodioloside), and andrographolide, have evidence of therapeutic efficacy from clinical human clinical studies [131,132,133,134,135,136,137]. It is remarkable that the purified constituents of adaptogenic extracts have different gene expression profiles than the extracts wherefrom there were isolated, and the number of deregulated genes is not correlated with the number of compounds in the extract, e.g., Herba Andrographis extract is a mixture of 39 constituents, such as andrographolides, flavonoids, etc., and deregulates 207 genes, while purified andrographolide affects the expressions of 626 genes [13].

## 4. Conclusions

This review provides a rationale for the pleiotropic therapeutic efficacy of adaptogens for the first time based on evidence from recent gene expression studies in target cells and where the network pharmacology and systems biology approaches were applied. The specific molecular targets and adaptive stress response signaling mechanisms involved in nonspecific modes of action of adaptogens are identified.

## Figures and Tables

**Figure 1 pharmaceuticals-15-01051-f001:**
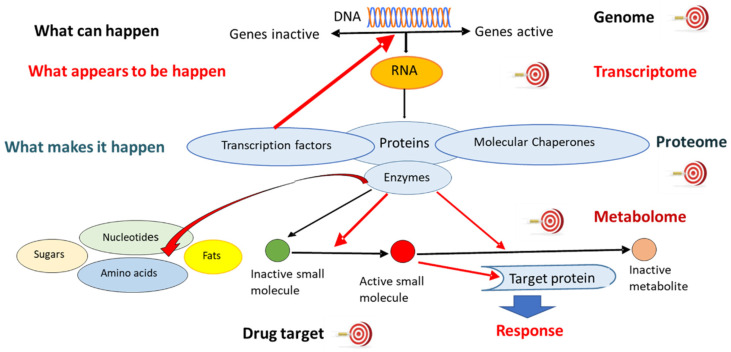
Flowchart showing the possible cellular and molecular targets for pharmacological intervention and the cell response after an active molecule binds its receptor at metabolomic, proteomic, transcriptomic, and genomic levels of regulation. Reprinted from Reference [5].

**Figure 2 pharmaceuticals-15-01051-f002:**
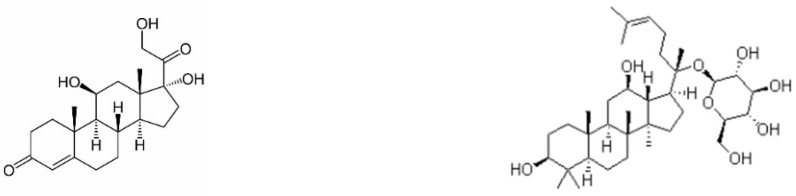
Chemical structure of the stress hormone cortisol and the Compound K [118], the primary active metabolite of ginsenosides.

**Figure 3 pharmaceuticals-15-01051-f003:**
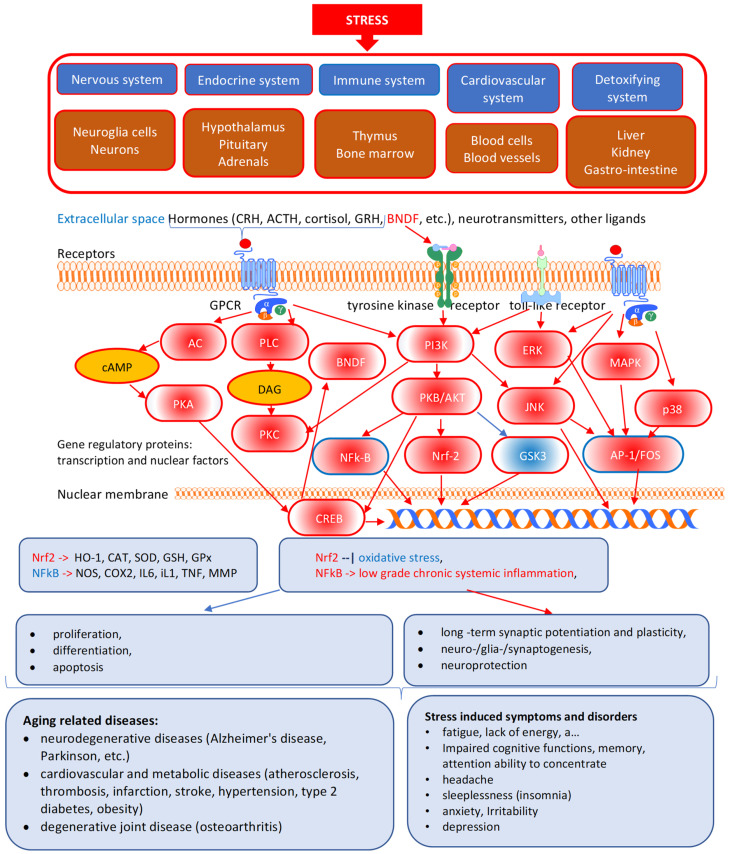
The molecular mechanisms and modes of the pharmacological action of red ginseng. Effects of red ginseng and ginsenosides on key mediators of neuro-endocrine immune complex, cardiovascular and detoxifying systems involved in the regulation of adaptive stress response to stressors/pathogens in stress and aging-induced diseases and disorders. CRH- and ACTH-induced stimulation of GPCR receptors activates the cAMP-dependent protein kinase (PKA) signaling pathway in the regulation of energy balance and metabolism across multiple systems, including adipose tissue (lipolysis), liver (gluconeogenesis, glucose tolerance), pancreases, and gut (insulin exocytosis and sensitivity), etc. The key molecules involved in the PI3K-Akt signaling pathway are receptor tyrosine kinase (RTKs). Activating the PI3K-Akt signaling pathway promotes cell proliferation and growth, stimulates cell cycle progression, metabolism, glycolysis, gluconeogenesis, proteins synthesis, energy storage, angiogenesis, vasodilatation, vascular remodeling, cell survival, and inhibits cell apoptosis in response to extracellular signals. Nonspecific antiviral action of ginseng is associated with activation of innate immunity by upregulation of the expression of the pathogen’s pattern recognition receptors, specifically toll-like receptors TLR-mediated signaling pathways. The protein kinase C (PKC) family of protein kinase enzymes with 15 isoforms plays an essential cell-type-specific role, particularly in the immune system through phosphorylation of CARD-CC family proteins and subsequent NF-κB activation. Three stress-activated MAPK signaling pathways playing important roles in cell proliferation, differentiation, survival, and death have been implicated in the pathogenesis of many human diseases, including Alzheimer’s disease, Parkinson’s disease, and cancer. (1) The stress factors inducing the activation of the c-Jun N-terminal kinase (JNK)/stress-activated protein kinase (SAPK) mediated adaptive signaling pathway are heat shock, irradiation, reactive oxygen species, cytotoxic drugs, inflammatory cytokines, hormones, growth factors, and other stresses. The activation of the JNK/MAPK10 signaling pathway promotes cell death and apoptosis via the upregulation of pro-apoptotic genes. (2) The activation of the extracellular-signal-regulated kinase (ERK) pathway is initiated by hormones and stresses to trigger endothelial cells proliferation during angiogenesis, T cell activation, long-term potentiation in hippocampal neurons, phosphorylation of the transcription factor p53, activation of phospholipase A2 in mast cells, followed by activation of biosynthesis leukotrienes and inflammation/allergy, etc. (3) The third major stress-activated p38 signaling pathway contributes to control of inflammation, the release of cytokines by macrophages and neutrophils, apoptosis, cell differentiation, and cell cycle regulation. Activation is shown in red, while the inhibition is in blue color cycles/ellipses (effect of ginseng/ginsenosides), arrows, and clouds. BDNF, brain-derived neurotrophic factor; cAMP, cyclic adenosine monophosphate; CREB, cAMP-responsive element-binding protein; ERK, extracellular signal-regulated kinase; GSK-3β, glycogen synthase kinase-3β; JNK; the c-Jun N-terminal kinase (JNK)/stress-activated protein kinase (SAPK MAPK, mitogen-activated protein kinase; NF-κB, nuclear factor-kappa B; Nrf2, nuclear factor E2-related factor 2; PI3K, phosphatidylinositol 3-kinase; PKA, protein kinase A; PKB, protein kinase B; PLC, phospholipase C.

**Figure 4 pharmaceuticals-15-01051-f004:**
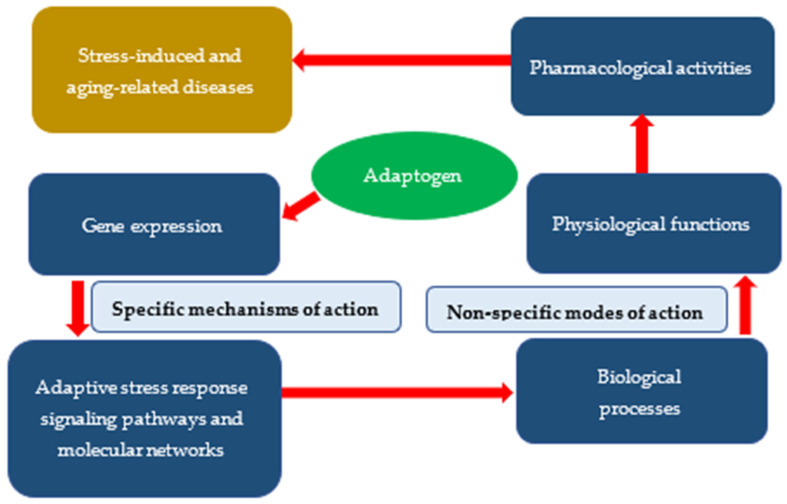
The rationale of specific and nonspecific pleiotropic actions of adaptogens.

**Figure 5 pharmaceuticals-15-01051-f005:**
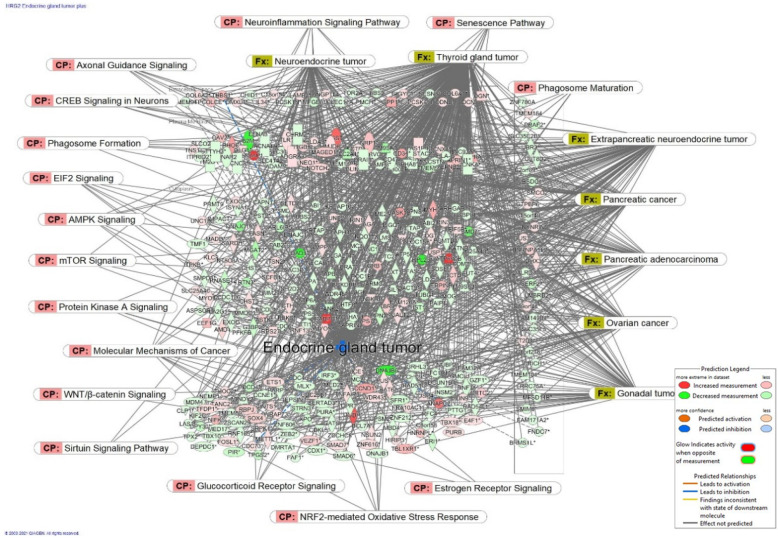
The molecular network shows predicted inhibition of exocrine gland tumor by an extract of red ginseng at a concentration of 1 μg/mL. Solid red or green color nodes indicate upregulated and downregulated genes, respectively; color intensity indicates the actual log-fold changes—the tags labeled with purple display the canonical pathways related to particular genes of the network. The titles marked with khaki show various types of tumors associated with the molecules in subnetworks. Reprinted from Reference [8].

**Table 1 pharmaceuticals-15-01051-t001:** Pharmacological activities of adaptogens. Adapted from [6,9,13,113].

Stimulating and tonic effects on the CNS system
Modulation of the stress response system, including the hypothalamus–hypophysis–adrenal (HPA) axis
Modulation of the endocrine system and metabolic regulation
Regulation of cellular homeostasis and metabolism
Modulation of the immune response
Increased expression of defensins peptides Increased expression of pathogen pattern recognition proteins-TLRs Increased expression of interferons Inhibition of cytokines release Inhibition of NF-κB Activation of natural killer cells Activation of phagocyting cells Activation of T- and B-lymphocytes Activation of melatonin signaling pathway
Anti-inflammatory activity
Inhibition of NF-κB-mediated signaling Inhibition of PLA2, arachidonic acid release, and metabolism Inhibition of nitric oxide generation
Detoxification and reparation of oxidative stress-induced damages in compromised cells
Activation of NRF2-signaling pathway proteins (KEAP1) Expression of phases I/II-metabolizing and antioxidant enzymes: glutathione S-transferase (GST), NAD(P)H quinone oxidoreductase 1 (NQO1), superoxide dismutase (SOD), and heme oxygenase 1 (HO1). Molecular chaperon (Hsp70)-mediated cytoprotection and repair processes Activation of melatonin signaling pathway
Direct antiviral activity
Inhibition of virus binding to host cell and its fission into the cytoplasm Termination of the viral life cycle in the infected host cell

**Table 2 pharmaceuticals-15-01051-t002:** Most essential genes regulated by adaptogens and associated signaling pathways, biological processes, physiological functions, and diseases; adapted from Reference [8].

Type(s)	Gene Symbol	Gene and Regulated Protein Name	Signaling Pathways	Biological Processes	Physiol. Functions	Diseases
Hormones	CRH	Corticotropin-releasing hormone	2	54	7	18
	* ACTH *	Adrenocorticotrophic hormone; ACTH	2			
	* UCN *	Urocortin (corticotropin-releasing factor family)	1	53	7	4
	* GNRH1 *	Gonadotropin-releasing hormone 1	3	25	10	12
Transmembrane receptors	* TLR9 *	Toll-like receptor 9, member of PI3K (complex)	152	65	7	66
	* CHRNE *	Cholinergic receptor nicotinic epsilon subunit	3	12	8	22
	* PRLR *	Prolactin receptor	2	17	7	11
G-protein coupled receptor	* CHRM4 *	Cholinergic receptor muscarinic 4	5	11	5	173
Nuclear receptor	* RORA *	RAR-related orphan receptor A (RZR)	Melatonin signaling	-	16	17
Transcription regulators	* STAT5A *	Signal transducer and activator of transcription 5A	19	57	13	10
	* FOS *	Fos proto-oncogene, AP-1 transcription factor subunit	21	37	15	52
	* FOXO6 *	Forkhead box O6	3	7	5	10
Kinases	* FLT1 *	Fms-related tyrosine kinase 1	9			
	*MAPK10* *JNK, SAPK1*	Mitogen-activated protein kinase 10, c-Jun N-terminal kinase	77	12	11	8
	*MAPK13* *p38, SAPK2*	Mitogen-activated protein kinase 13, p-38 MAP kinase	59	14	10	15
	* PRKCH *	Protein kinase C eta	72	15	11	20
	PKA	protein kinase A ACTH induced	cAMP/PKA/CREB signaling			
	* PKB *	Protein kinase B - AKT				
Metabolic enzymes	* GUCY1A2 *	Guanylate cyclase 1 soluble subunit alpha 2	19	4	6	32
	* HSPA6 *	Heat shock protein family A (Hsp70) member 6	6	3	5	3
	* PDE3B *	Phosphodiesterase 3B	16			
	* PDE9A *	Phosphodiesterase 9A	6			

Upregulated genes are shown in red, while downregulated genes are in blue.

**Table 3 pharmaceuticals-15-01051-t003:** The effects of adaptogens on canonical pathways are commonly involved in regulating adaptive stress response signaling; adapted from Reference [6].

Canonical Pathways
AMPK signaling Axonal guidance signaling Calcium signaling cAMP-mediated signaling Cardiac β-adrenergic signaling Chronic obstructive pulmonary disease signaling Colorectal cancer metastasis signaling Corticotropin-releasing hormone signaling CREB signaling in neurons CXCR4 signaling Dendritic cell maturation signaling Dopamine-DARPP32 feedback in cAMP signaling eNOS signaling Glutamate receptor signaling GP6 signaling pathway G-protein-coupled receptor signaling Inositol pyrophosphate biosynthesis Leptin signaling in obesity LPS-stimulated MAPK signaling Melatonin signaling and degradation Neuroinflammation signaling pathway Neuropathic pain signaling in dorsal horn neurons Nitric oxide signaling in the cardiovascular system NRF2-mediated oxidative stress response signaling Opioid signaling pathway Protein kinase A signaling Relaxin signaling Renin–angiotensin signaling Osteoblasts, osteoclasts, and chondrocytes in rheumatoid arthritis signaling Salvage pathways of pyrimidine nucleotide signaling Sperm motility signaling Super pathway of inositol phosphate compounds signaling Synaptic long-term depression signaling Telomere extension by telomerase signaling tRNA splicing signaling

**Table 4 pharmaceuticals-15-01051-t004:** Main cellular functions that are most influenced by adaptogens; adapted from Reference [15].

Cellular Function	Genes
**Cellular compromise:** − Oxidative stress response of blood cells − Degranulation of β-islet cells − Damage to mitochondria − Degeneration of hepatocytes − Cytotoxicity of cytotoxic T cells − Fragmentation of photoreceptor outer segments − Degeneration of retinal cone cells	*AIPL1*, *ALOX12*, *CDHR1*, *NGB3*, *GNLY*, *HLA-B*, *NCAM1*, *SERPINA1*, *ULBP3*, *XRCC5*
**Cell signaling**	*PDE3A, MUC20, PDE4D, PDE11A, ESR1, CCKBR*
**DNA replication, recombination, and repair**	*PARPBP*, *PDE3A*, *APLF*, *PDE4D*, *PDE11A*, *XRCC5*, *AICDA*
**Nucleic acid metabolism**	*PFKFB1*, *MTNR1A*, *PDE3A*, *APOBEC2*, *TAAR1*, *PDE4D*, *PDE11A*, *AIPL1*, *ESR1*, *AICDA*
**Lipid metabolism**	*NR4A3*, *RGS3*, *SLC27A2*, *AKR1D1*, *TNXB*, *SERPINA1*, *ALOX12*, *ESR1*, *CCKBR*, *CETP*, *NCAM1*

**Table 5 pharmaceuticals-15-01051-t005:** Age-associated diseases and genes involved in their pathogenesis and progression that are significantly regulated by adaptogens; downregulated genes are in blue and upregulated ones are in red colors; adapted from Reference [15].

Category	Diseases	Genes Affected by Adaptogens
Organismal injury and abnormalities	Physical disability Degeneration of retinal cone cells Atrophy of gastric mucosa Hypoestrogenism Postmenopausal vulvar atrophy Nociception Cone dystrophy Pelvic organ prolapse	* PDE11A,PDE3A,PDE4D * *AIPL1,CNGB3* * CCKBR * * ESR1 * *ESR1б MTNR1A* * KCNK10, PDE11A,PDE3A,PDE4D,SCN2B * *CDHR1,CNGB3* * ESR1, SERPINA1 *
Inflammatory and pulmonary diseases	Pulmonary emphysema Bronchiectasis Chronic bronchitis Chronic obstructive pulmonary disease	* PDE11A,PDE3A,PDE4D,SERPINA1 * * PDE11A,PDE3A,PDE4D * * MMP8,MTNR1A * * PDE11A,PDE3A,PDE4D,SERPINA1 *
Neurological and psychological diseases	Non-24 h sleep–wake disorder Sleep–wake schedule disorder	* MTNR1A * * PDE3A *
Cardiovascular diseases	Ischemic cardiomyopathy Cholesteryl ester transfer protein deficiency Angina pectoris Cerebral small vessel disease	* PDE11A,PDE3A,PDE4D,PPP1R1A * * CETP * * PDE11A,PDE3A,PDE4D–all upregulated * * PDE3A-unregulated *
Skeletal and connective tissues	Osteochondrodysplasia	*COL9A1, PDE4D*
Metabolic disease	Estrogen resistance	* ESR1 *

## Data Availability

Data sharing not applicable.

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
