# Peer review of "Network Pharmacology of Adaptogens in the Assessment of Their Pleiotropic Therapeutic Activity"

_pharmaceuticals, 2022, doi:10.3390/ph15091051_

Round 1

Reviewer 1 Report

The manuscript “Network pharmacology of adaptogens in the assessment of their pleiotropic therapeutic activity“ is interesting and well written. The manuscript discussed a rationale for the pleiotropic therapeutic efficacy of adaptogens based on evidence from recent gene expression studies in target cells and where the network pharmacology and systems biology approach was applied.

. However, I have some suggestions that might improve the manuscript:

- The introduction needs to be further extended to give the readers a brief background about a number of important concepts e.g. adaptogens, network pharmacology, ligand-receptor concept vs polyvalent pharmacology concept.

- It will be very useful to add the limitations of using network pharmacology in drug development and the use of crude extracts as therapeutic agents at the end of your manuscript.

- Can a single compound act as an adaptogen, or this is restricted only to complex extracts? Please discuss this point revealing that even pure single compound-based drugs may also act as adaptogens.

Author Response

We thank the reviewer for thoroughly reviewing the manuscript and for thoughtful criticism. We have revised the manuscript to address the comments and followed all suggestions to strengthen the manuscript. Detailed responses to the reviewer's comments are shown below point by point in red color text.

General comments:

Point 1: The introduction needs to be further extended to give the readers a brief background about a number of important concepts, e.g., adaptogens, network pharmacology, ligand-receptor concept vs polyvalent pharmacology concept. 

Response 1: The comment made by the reviewer was addressed in the introduction

Pharmacology concerns drug action on physiological systems for therapeutic benefit, focusing on theories, procedures, and mechanisms related to the chemical control of physiological processes.

Pharmacology aims to define the molecular events initiating drug effects at the pharmacological targets in therapeutic and other systems. The term "pharmacological target" refers to the biomolecules, such as DNA, mRNA, and proteins, including transmembrane and nuclear receptors, ion channels,  transport proteins, and numerous enzymes to which a drug binds first in the body to elicit its pharmacologic effect (Figure 1).

The complementary binding of drug molecules to a protein with a physiological purpose in the cell can change physiological response and result in a pharmacologic effect. The efficacy of binding a drug  (ligand) to a specific biomolecule (receptor) depends on the drug's chemical structure and affinity.

Historically, the drug development strategy was based on the assumption that a single target mechanism of action is the best option to obtain the target-specific therapeutic, which is selective for treating specific conditions and free of adverse events. However, in reality, many drugs and natural compounds interact and bind with multiple receptors (multitarget interaction), resulting in polyvalent pharmacological action and pleiotropic therapeutic activity. For example, the polytropic therapeutic activity of ginseng and some natural compounds or "Ginseng-like" plant extracts, collectively known as adaptogens, is associated with their polyvalent mode of action on the neuroendocrine-immune complex (stress system) [5, 7, 8] and multitarget effects on adaptive stress response signaling pathways and molecular networks allied with these pharmacological targets [15]. Initially, adaptogens were defined as "Ginseng-like" plants, which increase the so-called "state of nonspecific resistance" of an organism to stress, resulting in tonic, stress-protective, and adaptogenic activity [95, 110, 114]. Adaptogens are currently defined as a therapeutic category/pharmacological group of herbal medicines or/and nutritional products, increasing adaptability, survival, and resilience in stress and aging [15].

Point 2: It will be very useful to add the limitations of using network pharmacology in drug development and the use of crude extracts as therapeutic agents at the end of your manuscript.

Response 2: The comment made by the reviewer was adequately addressed in the revised version. There are several limitations of network pharmacology predictions based on in vitro and in silico studies, which must be further verified in animal experiments. The second limitation is the limited scientific information related to the direction of correlations between gene expression and physiological function or disease that is used in silico analysis for predictions of therapeutic efficacy or toxicity. Additional studies where different experimental outcome measures will be applied are required. Furthermore, a strong resistance mechanism can override all other effects of a drug. For instance, the drug efflux transporter P-glycoprotein in the cell membrane can expel drugs before they can reach their actual intracellular targets and thereby prevent all or most further downstream signaling and network effects. Lastly, clinical studies on predicted diseases in human subjects are essential.

According to European drug regulations  (EMA guidelines), the term "crude extract" is not used in the European Pharmacopeia and scientific literature. Instead, three types of herbal drug extracts are specified: standardized, quantified, and other extracts, where the content of a genuine extract is adjusted to the defined content or range of the active constituents with known therapeutic efficacy or to the content of the analytical markers. There are no limitations concerning pharmaceutical-grade herbal extracts of reproducible quality and pharmacological activity compared to purified active constituents since the content of a genuine extract is adjusted to the defined content or range of the active constituents with known therapeutic efficacy or to the content of the analytical markers

Point 3: Can a single compound act as an adaptogen, or this is restricted only to complex extracts? Please discuss this point, revealing that even pure single compound-based drugs may also act as adaptogens.

Response 3: Yes, purified compounds can also act as adaptogens [128,129] if there is evidence that they increase adaptability, survival, and resilience in stress and aging by triggering intracellular and extracellular adaptive signaling pathways of cellular and organismal defense systems, e.g., the generation of hormones (cortisol, corticotropin‐releasing hormone, and gonadotropin‐releasing hormones, urocortin, neuropeptide Y [15]. Currently, at least three purified adaptogens, ginsenoside Rg1, salidroside (rhodioloside), and andrographolide have evidence of therapeutic efficacy from clinical human clinical studies [130-136]. It is remarkable that the purified constituents of adaptogenic extracts have different gene expression profiles than the extracts wherefrom there were isolated, and the number of deregulated genes is not correlated with the number of compounds in the extract, e.g., Herba Andrographis extract is a mixture of 39 constituents such as andrographolides, flavonoids, etc. and deregulates 207 genes, while purified andrographolide affects the expression of 626 genes [12].

Todorova, V. ; Ivanov, K.; Ivanova, S. Comparison between the Biological Active Compounds in Plants with Adaptogenic Properties (Rhaponticum carthamoides, Lepidium meyenii, Eleutherococcus senticosus and Panax ginseng). Plants (Basel). 2021 Dec 26;11(1):64. doi: 10.3390/plants11010064. PMID: 35009068; PMCID: PMC8747685.

Todorova, V.; Ivanov, K.; Delattre, C.; Nalbantova, V.; Karcheva-Bahchevanska, D.; Ivanova, S. Plant Adaptogens-History and Future Perspectives. Nutrients. 2021 Aug 20;13(8):2861. doi: 10.3390/nu13082861. PMID: 34445021; PMCID: PMC8398443.

Lee, T.X.Y.; Wu, J. ; Jean, W.H. ;  Condello, G. ;  Alkhatib, A. ;  Hsieh, C.C. ;  Hsieh, Y.W. ;  Huang, C.Y. ; Kuo, C.H. Reduced stem cell aging in exercised human skeletal muscle is enhanced by ginsenoside Rg1. Aging (Albany NY). 2021 Jun 28;13(12):16567-16576. doi: 10.18632/aging.203176. Epub 2021 Jun 28. PMID: 34181580 Free PMC article. Clinical Trial. 

Wu, J. ;  Saovieng, S. ;  Cheng, I.S. ;  Liu, T. ;  Hong, S. ;  Lin, C.Y. ;  Su, I.C. ;  Huang, C.Y. ;  Kuo, C.H. Ginsenoside Rg1 supplementation clears senescence-associated β-galactosidase in exercising human skeletal muscle. J Ginseng Res. 2019 Oct;43(4):580-588. doi: 10.1016/j.jgr.2018.06.002. Epub 2018 Jun 21. PMID: 31695564 Free PMC article. 

Hou, C.W. ;  Lee, S.D. ;  Kao, C.L. ;  Cheng, I.S. ;  Lin, Y.N. ;  Chuang, S.J. ;  Chen, C.Y. ; Ivy, J.L. ;  Huang, C.Y. ;  Kuo, C.H. Improved inflammatory balance of human skeletal muscle during exercise after supplementations of the ginseng-based steroid Rg1. PLoS One. 2015 Jan 24;10(1):e0116387. doi: 10.1371/journal.pone.0116387. eCollection 2015. PMID: 25617625 Free PMC article. Clinical Trial.

Zhang, H. ;  Shen,W.S. ;  Gao, C.H. ;  Deng, L.C. ;  Shen, D. Protective effects of salidroside on epirubicin-induced early left ventricular regional systolic dysfunction in patients with breast cancer. Drugs R D. 2012 Jun 1;12(2):101-6. doi: 10.2165/11632530-000000000-00000. PMID: 22770377; PMCID: PMC3585960.

Panossian, A. ;  Wikman, G. ;  Sarris, J. Rosenroot (Rhodiola rosea): traditional use, chemical composition pharmacology and clinical efficacy. Phytomedicine. 2010 Jun;17(7):481-93. doi: 10.1016/j.phymed.2010.02.002. Epub 2010 Apr 7. PMID: 20378318.

Aksenova, R.A. ;  Zotova, M.I. ;  Nekhoda, M.F. ;  Cherdintsev, S.G. 1968. Comparative characteristics of the stimulating and adaptogenic effects of Rhodiola rosea preparations. In: Saratikov, A.S. (Ed.), Stimulants of the Central Nervous System, vol. 2. Tomsk University Press, Tomsk, pp. 3–12

Ciampi, E. ;  Uribe-San-Martin, R. ;  Cárcamo, C. ;  Cruz, J.P. ;  Reyes, A. ; Reyes, D. ;  Pinto, C. ;  Vásquez, M. ; Burgos, R.A. ;  Hancke, J. Efficacy of andrographolide in not active progressive multiple sclerosis: a prospective exploratory double-blind, parallel-group, randomized, placebo-controlled trial. BMC Neurol. 2020 May 7;20(1):173. doi: 10.1186/s12883-020-01745-w. PMID: 32380977; PMCID: PMC7203851.

We are grateful to the reviewer for some of his valuable comments, which helped us to improve the quality of this manuscript. We hope that the clarifications above and revisions made can be helpful for your decision, and the revised manuscript will be recommended for publication.    

Reviewer 2 Report

Review Comments: pharmaceuticals-1808970-R1

Comments to Authors

Panossian et al. Review article "Network pharmacology of adaptogens in the assessment of their pleiotropic therapeutic activity" As shown in the manuscript, the authors demonstrate that the reductionist concept, based on ligand-receptor interaction, is not a suitable model for adaptogens, and herbal preparations affect multiple physiological functions, revealing polyvalent pharmacological activities and are traditionally used in many conditions. Overall, this manuscript described the specific molecular targets and adaptive stress response signaling mechanisms involved in nonspecific modes of action of adaptogens. However, it was still some viewpoints in this manuscript that must be noted.

Comments

1.    In 2.2.3. Proteomics: Proteomic methods the expression of all proteins including isoforms. It was well known that the naturally occurring form of glucose is d-glucose, while l-glucose is produced synthetically in comparatively small amounts and is less biologically active. Moreover, in line 368: The ginsenosides act primarily on the hypothalamus and pituitary, stimulating ACTH secretion, However, line 372 described that Ginsenoside Rd, inhibits corticosterone secretion in the cells and inhibits ACTH-induced corticosterone biosynthesis. Moreover, just like the authors described the specific molecular targets and adaptive stress response (agonist) signaling mechanisms involved in nonspecific modes or antagonists of action of adaptogens. That seems that we could not identify from the Network pharmacology of adaptogens in the assessment of their pleiotropic therapeutic activity. Suggest the authors must change the title to avoid the readers get the confusion from the title could be better.

2.    The drugs (too many compounds that we can imagine them) could be binding to the protein (such as the receptors, enzymes, and ionic channels at metabolomic, proteomic, transcriptomic, and genomic levels of regulation (upregulation, downregulated, or non-regulation). Moreover, how those drugs could be absorbed, distributed, and other pharmacokinetic and pharmacodynamic (receptor binding and the show the differ organic effects) problems still should be faced. In this manuscript, we cannot find any novel theory. In Figure 1, we found the authors too simpling the chart and described no at the complex. Suggest the authors must describe them well to avoid the readers get the confusion could be better. Like as the sugars, nucleotides, and amino acids could not pass through the cell membrane only activated on the membrane receptors. The action could be a different degree of activation, inhibition, or non-activation. They were too complicated, but the authors did not describe them well.

3.    In Figure 2, all the charts showed too simplifying and errors. The receptors of hormones are always in the cytosol. The nucleus transcript protein includes NF-κB, Nrf2, c-jun, and c-fos. Some are stimulated, same are inhibited or non-activation. But the authors made a lot of errors, making the chard dot worth citing. We can not find any good organization in this chart. Suggest the authors must re-paint the chard again and avoid the readers getting confused from though that chart could be better.

4.   What is the Ginseng HRG80 TM? The authors did not describe it. It could be a commercial activity. Moreover, we can not find any suggestions for the chart in Figure 4. Same as comment 1, it could not be a pure compound. The results could be so complicated (some stimulate some inhibited, and some did not have any action. Moreover, The different concentrations could produce different results. What do the authors want to show?  Suggest the authors must re-paint the chart again and avoid the readers getting confusion from this chard could be better.

5.    In precision pharmacology or targeted therapy. The docking affinity between the drugs and receptors is very important. Using the network pharmacology of adaptogens in the assessment of their pleiotropic therapeutic activity always discussing the pathways seems unreasonable. Suggest the authors explain them well and avoid the readers getting confusion from this manuscript could be better.

6.    This manuscript showed too many Tables and could be too complicated and too many signaling pathways. We can not get the key points that the authors want to say. Suggest the authors summarize them well and avoid the readers getting confusion from this manuscript could be better.

Reviewer 3 Report

1.      Page 8. Line 343. The authors mentioned the structure similarity of ginsenosides with cortisol. I would suggest that the authors add the general structure(s) of the ginsenosides and cortisol to show the similarity.

2.      Page 11. Line 461. Supplement 1 was not found in the manuscript. We cannot see Table 6 and Figure 3 in detail.

3.      Page 12. Line 470. Figure 4. (a) Is the Ginseng HRG80 final product or crude extract? What is the purity of Ginseng HRG80? Did the authors perform the HPLC experiment to analyze the profile of the testing sample? (b) I would suggest that the authors briefly describe the molecular network data processing.

4.      Page 12-16. In Table 1-5, add a column for the citation of references.

Author Response

We thank the reviewer for thoroughly reviewing the manuscript and for thoughtful criticism. We have revised the manuscript to address the comments and followed all suggestions to strengthen the manuscript. Detailed responses to the reviewer's comments are shown below point by point in red color text.

Comment 1:    Page 8. Line 343. The authors mentioned the structure similarity of ginsenosides with cortisol. I would suggest that the authors add the general structure(s) of the ginsenosides and cortisol to show the similarity.

Response 1: The comment made by the reviewer was adequately addressed in the revised version.

Figure 2 shows the chemical structures of stress hormone cortisol and the Compound K, the primary active metabolite of ginsenosides

Comment 2:   Page 11. Line 461. Supplement 1 was not found in the manuscript. We cannot see Table 6 and Figure 3 in detail.

Response 1: The comment made by the reviewer was adequately addressed in the revised version.

The number of figures and tables was different in several drafts of this review resulting in omissions in the submitted version.

Comment 3:   Page 12. Line 470. Figure 4. (a) Is the Ginseng HRG80 final product or crude extract? What is the purity of Ginseng HRG80? Did the authors perform the HPLC experiment to analyze the profile of the testing sample? (b) I would suggest that the authors briefly describe the molecular network data processing.

Figure 4 shows an example of a molecular network and numerous interactions associated with predicted inhibition of exocrine gland tumor by an extract of Red Ginseng".

For details of this study, we refer to original publication, 

Panossian, A.; Abdelfatah, S.; Efferth, T. Network pharmacology of ginseng (part II): The differential effects of red ginseng and ginsenoside Rg5 in cancer and heart diseases as determined by transcriptomics. Pharmaceuticals 2021, 14(10), 1010; doi.org/10.3390/ph14101010.

wherefrom a curious reader can find all details related to the Red ginseng preparation in the section 4.1. (below) including HPLC fingerprint and content of ginsenosides Rg1, Re, Rf, Rb1, Rg2, Rc, Rh1, Rb2, F1, Rd, Rg6, F2, Rh4, Rg3-(S-R), PPT (20-R), Rk1, C(k), Rg5, Rh2, Rh3, 20S-PPT, and PPD in the supplements)

Powdered Red Ginseng preparation HRG80TM was manufactured at Botalys S. A. (Ath, Belgium).

Korean Ginseng (P. ginseng Meyer) root was hydroponically cultivated in controlled conditions, air-dried, steamed to Red Ginseng, which was powdered, and standardized for the content of ginsenosides Rg1, Re, Rf, Rb1, Rg2, Rc, Rh1, Rb2, F1, Rd, Rg6, F2, Rh4, Rg3-(S-R), PPT (20-R), Rk1, C(k), Rg5, Rh2, Rh3, 20S-PPT, and PPD to obtain Red Ginseng HRG80TM preparation containing 7.6% of total ginsenosides (Supplemental Table S4 in Supplement 11). HRG80TM preparation was exhaustively extracted by 40% ethanol and evaporated to dryness to obtain HRG extract (DER 4: 1) which was used in vitro experiments. The content of ginsenosides in HRG extract was 30.32% (Supplement 11). 

The reference standard, P. ginseng Meyer powdered root preparation, and the extract contained 5.57% and 22.15% total ginsenosides correspondingly (Supplemental Table S4 in Supplement 11). All herbal preparations were analyzed and certified by Botalys S. A.

Comment 4:         Page 12-16. In Table 1-5, add a column for the citation of references.

Response 4: The comment made by the reviewer was addressed in the introduction.

We are grateful to the reviewer for his valuable comments, which helped us to improve the quality of this manuscript. We hope that the clarifications above and revisions made can be helpful for your decision, and the revised manuscript will be recommended for publication.   

Round 2

Reviewer 2 Report

Review Comments: pharmaceuticals-1808970-R2

Comments to Authors

Panossian et al. review the article "Network pharmacology of adaptogens in the assessment of their pleiotropic therapeutic activity" As shown in the manuscript, the authors demonstrate that based ligand-receptor interaction, is not a suitable model for adaptogens, and herbal preparations affect multiple physiological functions, revealing polyvalent pharmacological activities and are traditionally used in many conditions. Overall, this manuscript described The specific molecular targets and adaptive stress response signaling mechanisms involved in nonspecific modes of action of adaptogens are identified. However, we just did not understand that the manuscripts all described the effects of ginsenosides but did not show them in the title and abstract. Moreover, this manuscript must prevent self-plagiarism.

Comments

1.    Figure 1 was the same as Figure 7 of DOI:10.1111/nyas.13399 (reference 5).

2.    The compound K in Figure 2, we can not find described in this manuscript.

3.    Figure 3 was the same as Figure 8 of PMID: 30466987 (reference 6). It did not describe the molecular mechanisms and modes of the pharmacological action of Red Ginseng. The authors only seem to show the ginsenoside as the stress response but did explain or supplement enough evidence.

4.    Figure 5 was the same as Figure 6 of PMID: 34681234 (reference 7).

5.    Table 2 same as Table 1 of PMID: 30466987 (reference 8).

6.    Table 3 same and only rearrange from Table 3 of PMID: 30466987 (reference 8).

7.    Table 5 same as Table 10 of PMID: 33103257 (reference 15).

Round 3

Reviewer 2 Report

Review Comments of pharmaceuticals-1808970-R3

Response to Reviewer 2 Comments R2

We thank the reviewer for thoroughly reviewing the manuscript and for thoughtful criticism. We have revised the manuscript to address the comments and followed all suggestions to strengthen the manuscript (in red color text. Detailed responses to the reviewer's comments are shown below point by point.

Point 1a: Panossian et al. review the article "Network pharmacology of adaptogens in the assessment of their pleiotropic therapeutic activity" As shown in the manuscript, the authors demonstrate that based ligand-receptor interaction, is not a suitable model for adaptogens, and herbal preparations affect multiple physiological functions, revealing polyvalent pharmacological activities and are traditionally used in many conditions. Overall, this manuscript described The specific molecular targets and adaptive stress response signaling mechanisms involved in nonspecific modes of action of adaptogens are identified. However, we just did not understand that the manuscripts all described the effects of ginsenosides but did not show them in the title and abstract.

Response 1a: The manuscript describes the effects of various adaptogens, including Panax ginseng C.A. Mayer, Eleutherococcus senticosus (Rupr. and Maxim.) Maxim., Rhodiola rosea L., Withania somnifera (L.) Dunal., Schisandra chinensis (Turcz.) Baill. Andrographis paniculata (Burm. f.) Nees , Rhaponticum cartamoides Iljin, and their active constituents including ginsenosides, salidroside, rosavins, schisandras, eleutherosides, withanolides and andrographolides. The review is not exclusively about ginsenosides in order to specify them in the title.    

Comment from the reviewer for Response to Point 1a:

The authors only self-cited 3 papers and still did not cite any other authors’ references. Suggesting the authors must cite them well, supplement enough evidence, and explain well in the manuscript.

Point 1b: Moreover, this manuscript must prevent self-plagiarism.

1. Figure 1 was the same as Figure 7 of DOI:10.1111/nyas.13399 (reference 5).

2. The compound K in Figure 2, we cannot find described in this manuscript.

3. Figure 5 was the same as Figure 6 of PMID: 34681234 (reference 7).

4. Table 2 same as Table 1 of PMID: 30466987 (reference 8).

5. Table 3 same and only rearrange from Table 3 of PMID: 30466987 (reference 8).

Response 1b: We appreciate the reviewer's comment, but it seems that he misunderstood or misinterpreted the journal policy related to self-plagiarism - that is when the author copies text from his own previously published articles without copyrights from the journal where it has been published.

In this review, we provide some figures and tables from our previous publications having copyrights of the publisher. That is not a copyright infringement and violation of journal rules; there is nothing about that the authors are not allowed to copy our copyright tables and figures.

We used our original publications related to different topics as supportive material to justify this review's new claims.

Comment from the reviewer for Response to Point 1b:

A.     According to the “Instructions for Authors” of Pharmaceuticals,

Plagiarism includes copying text, ideas, images, or data from another source, even from your publications, without giving any credit to the original source.

B.     According to the “Copyrights” of MPDI,

Reproducing Published Material from other Publishers

It is absolutely essential that authors obtain permission to reproduce any published material (figures, schemes, tables or any extract of a text) which does not fall into the public domain, or for which they do not hold the copyright. Permission should be requested by the authors from the copyright holder (usually the Publisher, please refer to the imprint of the individual publications to identify the copyright holder).

Permission is required for:

1.       Your own works published by other Publishers and for which you did not retain copyright.

2.       Substantial extracts from anyone's works or a series of works.

3.     Use of Tables, Graphs, Charts, Schemes and Artworks if they are unaltered or slightly modified.

4.       Photographs for which you do not hold copyright.

The authors must avoid using a lot of the same or similar Figures or Tables in the different papers.

1.      The authors must avoid using text, ideas, images, or data from their publications, even with (such as one manuscript, but submitted to two different journals) or without giving any credit to the original source. 

2.      Although ginsenosides could be part of the adaptogens, the Figures or Tables still do not represent all the adaptogens. The authors must avoid copying and pasting the same figure of ginsenoside represents all the adaptogens but did not supplement enough evidence or cited the other author’s reference.

3.      The authors must prevent self-plagiarism. Suggest the authors must cite the other authors’ references to redraw the related Figures and cite the other authors’ references to rewrite the related Tables, supplement enough evidence and explain well the above five comments point-by-point response to the Reviewer’s comments.

Point 2: The compound K in Figure 2, we can not find described in this manuscript.

Response 2: Ginsenoside CK is the primary active metabolite of ginsenosides and represents basic chemical structure of ginsenosides as it is mentioned in the figure legend 2. Chemical structure of stress hormone cortisol and the Compound K, the primary active metabolite of ginsenosides.

Comment from the reviewer for Response to Point 2:

The Figure legend only described the detailed information of the Figure. However, the authors still did not describe the role of compound K and explain anything in this manuscript. Suggest the authors must describe the role of compound K, cite the references, supplement enough evidence and explain well the compound K in the manuscript.

Round 4

Reviewer 2 Report

Review Comments of pharmaceuticals-1808970-R3

Comments to Authors

Panossian et al. review the article "Network pharmacology of adaptogens in the assessment of their pleiotropic therapeutic activity" As shown in the manuscript, the authors demonstrate that based ligand-receptor interaction, is not a suitable model for adaptogens, and herbal preparations affect multiple physiological functions, revealing polyvalent pharmacological activities and are traditionally used in many conditions. Overall, this manuscript described The specific molecular targets and adaptive stress response signaling mechanisms involved in nonspecific modes of action of adaptogens are identified. However, we just did not understand that the manuscripts all described the effects of ginsenosides but did not show them in the title and abstract. Moreover, this manuscript must prevent self-plagiarism.

Comment from the reviewer

Point 1a: Panossian et al. review the article "Network pharmacology of adaptogens in the assessment of their pleiotropic therapeutic activity" As shown in the manuscript, the authors demonstrate that based ligand-receptor interaction, is not a suitable model for adaptogens, and herbal preparations affect multiple physiological functions, revealing polyvalent pharmacological activities and are traditionally used in many conditions. Overall, this manuscript described The specific molecular targets and adaptive stress response signaling mechanisms involved in nonspecific modes of action of adaptogens are identified. However, we just did not understand that the manuscripts all described the effects of ginsenosides but did not show them in the title and abstract.

Response 1a: The manuscript describes the effects of various adaptogens, including Panax ginseng C.A. Mayer, Eleutherococcus senticosus (Rupr. and Maxim.) Maxim., Rhodiola rosea L., Withania somnifera (L.) Dunal., Schisandra chinensis (Turcz.) Baill. Andrographis paniculata (Burm. f.) Nees , Rhaponticum cartamoides Iljin, and their active constituents including ginsenosides, salidroside, rosavins, schisandras, eleutherosides, withanolides and andrographolides. The review is not exclusively about ginsenosides in order to specify them in the title.    

Comment from the reviewer for Response to Point 1a:

The authors only self-cited 3 papers and still did not cite any other authors’ references. Suggesting the authors must cite them well, supplement enough evidence, and explain well in the manuscript.

Point 1b: Moreover, this manuscript must prevent self-plagiarism.

1. Figure 1 was the same as Figure 7 of DOI:10.1111/nyas.13399 (reference 5).

2. The compound K in Figure 2, we cannot find described in this manuscript.

3. Figure 5 was the same as Figure 6 of PMID: 34681234 (reference 7).

4. Table 2 same as Table 1 of PMID: 30466987 (reference 8).

5. Table 3 same and only rearrange from Table 3 of PMID: 30466987 (reference 8).

Response 1b: We appreciate the reviewer's comment, but it seems that he misunderstood or misinterpreted the journal policy related to self-plagiarism - that is when the author copies text from his own previously published articles without copyrights from the journal where it has been published.

In this review, we provide some figures and tables from our previous publications having copyrights of the publisher. That is not a copyright infringement and violation of journal rules; there is nothing about that the authors are not allowed to copy our copyright tables and figures.

We used our original publications related to different topics as supportive material to justify this review's new claims.

Comment from the reviewer for Response to Point 1b:

A.     According to the “Instructions for Authors” of Pharmaceuticals,

Plagiarism includes copying text, ideas, images, or data from another source, even from your publications, without giving any credit to the original source.

B.     According to the “Copyrights” of MPDI,

Reproducing Published Material from other Publishers

It is absolutely essential that authors obtain permission to reproduce any published material (figures, schemes, tables or any extract of a text) which does not fall into the public domain, or for which they do not hold the copyright. Permission should be requested by the authors from the copyright holder (usually the Publisher, please refer to the imprint of the individual publications to identify the copyright holder).

Permission is required for:

1.       Your own works published by other Publishers and for which you did not retain copyright.

2.       Substantial extracts from anyone's works or a series of works.

3.     Use of Tables, Graphs, Charts, Schemes and Artworks if they are unaltered or slightly modified.

4.       Photographs for which you do not hold copyright.

The authors must avoid using a lot of the same or similar Figures or Tables in the different papers.

1.      The authors must avoid using text, ideas, images, or data from their publications, even with (such as one manuscript, but submitted to two different journals) or without giving any credit to the original source. 

2.      Although ginsenosides could be part of the adaptogens, the Figures or Tables still do not represent all the adaptogens. The authors must avoid copying and pasting the same figure of ginsenoside represents all the adaptogens but did not supplement enough evidence or cited the other author’s reference.

3.      Although all articles are published in MDPI journals, copyright is retained by the authors. Articles are licensed under an open access Creative Commons CC BY 4.0 license, Permission is required for use of Tables, Graphs, Charts, Schemes, and Artworks if they are unaltered or slightly modified. The authors must prevent self-plagiarism. Suggest the authors must cite the other authors’ references to redraw the related Figures and cite the other authors’ references to rewrite the related Tables, supplement enough evidence and explain well the above five comments point-by-point response to the Reviewer’s comments.

Point 2: The compound K in Figure 2, we can not find described in this manuscript.

Response 2: Ginsenoside CK is the primary active metabolite of ginsenosides and represents basic chemical structure of ginsenosides as it is mentioned in the figure legend 2. Chemical structure of stress hormone cortisol and the Compound K, the primary active metabolite of ginsenosides.

Comment from the reviewer for Response to Point 2:

The Figure legend only described the detailed information of the Figure. However, the authors still did not describe the role of compound K and explain anything in this manuscript. Suggest the authors must describe the role of compound K, cite the references, supplement enough evidence and explain well the compound K in the manuscript.
